# Primitive Reflex Activity in Relation to Motor Skills in Healthy Preschool Children

**DOI:** 10.3390/brainsci11080967

**Published:** 2021-07-23

**Authors:** Anna Pecuch, Ewa Gieysztor, Ewelina Wolańska, Marlena Telenga, Małgorzata Paprocka-Borowicz

**Affiliations:** 1Physiotherapy Department, Faculty of Health Sciences, Wrocław Medical University, Grunwaldzka 2, 50-355 Wrocław, Poland; anna.pecuch@student.umed.wroc.pl (A.P.); marlena.telenga@student.umed.wroc.pl (M.T.); malgorzata.paprocka-borowicz@umed.wroc.pl (M.P.-B.); 2Department of Pediatrics, Division of Pediatrics and Rare Disorders, Wrocław Medical University, ul. Bartla 5, 51-618 Wroclaw, Poland; ewelina.wolanska@student.umed.wroc.pl

**Keywords:** primitive reflexes, preschool children, neuromotor maturity, physical development

## Abstract

Psychomotor development in the first year of life is possible due to activity and then integration of primitive (neonatal) reflexes. The presence of active primitive reflexes (APRs) in preschool and school-aged children indicates neuromotor immaturity. Studies show dependencies between the preserved activity of primary reflexes and developmental problems such as learning difficulties (problems with reading, writing, reduced mathematics skills, and dyslexia), difficulties with coordination, and attention deficit. The primary purpose of this study is to present the activity of three tonic reflexes in a sample of 112 Polish children aged 4–6 in relation to their motor skills. The children were examined for the presence of the asymmetric tonic neck reflex (ATNR), symmetric tonic neck reflex (STNR), and tonic labyrinthine reflex (TLR). Motor performance was examined with the MOT 4–6. Statistical analysis shows an inverse correlation between the score in the test of reflexes and motor efficiency (MOT 4–6) at *p* < 0.05 (−0.33). Children with increased reflex activity presented a lower level of motor efficiency. The multiple regression model showed that with the older age of the child and the decrease in the level of reflex activity, the motor skills of children improve. Thus, there is a need for early screening of primitive reflexes in children. Properly selected exercises and therapeutic activities aimed at integrating APRs in children with developmental difficulties can improve their motor skills, perceptual abilities, and emotional behavior.

## 1. Introduction

The first year of life is a time of intensive psychomotor development. The baby acquires motor skills such as stable lying on the back/belly accompanied by the development of eye–hand coordination, supporting oneself on forearms/hands, or quadrupling. These skills are made possible by the preceding activity of neonatal reflexes and their subsequent gradual integration [1]. Primary reflexes are stereotypical, and the brainstem governs involuntary motor reactions. They develop during fetal life and are strengthened postnatally (during childbirth and a few weeks afterward). They help function and interact with the environment in the first months of life [2,3]. After fulfilling their function, they are integrated (inhibited) through the emerging cortically directed volitional movements (there is a process of continuous myelination and maturation of connections with higher brain centers) [2,4]. Reflexes are replaced by more mature postural reactions that we use throughout our lives. Well-developed postural responses provide the basis for good balance, posture, and coordination [5]. The proper dynamics of functional maturation of the central nervous system enable the acquisition of higher-level cognitive skills like purposeful use of objects, learning to function independently, and participate in education and social activities [1,4,5]. Both the motor performance of a preschool child and the integration of primary reflexes are indicators (markers) of functional neuromotor maturity [5,6].

Primary reflexes include tonic reflexes. Particularly noteworthy are the asymmetrical tonic neck reflex (ATNR), the symmetrical tonic neck reflex (STNR), and the tonic labyrinthine reflex (TLR). All of them are activated by a tonic stimulus induced by the head and neck movement. ATNR is formed in the 18th week of fetal life; it is integrated between 3 to 9 months after birth [5,6]. The lateral head rotation provokes an extension of the upper and lower limbs on the facial side and a deflection on the occipital side. It is an activity that is the basis for the formation of eye–hand coordination. Its integration begins with gradual improvement of the work of the centerline of the body. The consequence of incomplete integration of ATNR might be poor eye–hand coordination, difficulties in crossing the visual midline. In the educational process, it can interfere with learning to read and cause orientation problems. In motor development, improper posture can be observed while walking [7]. The formation of scoliosis is also possible [8]. STNR emerges later, i.e., 6 to 8 months after birth, and integrates 9 to 11 months after birth. STNR flexion is caused by the flexion of the head, accompanied by the elevation of the pelvis with the simultaneous extension of lower limbs and flexion of upper limbs. STNR extension is caused by the extension of the head, accompanied by the extension of upper limbs and the flexion of lower limbs. At 6 to 9 months after birth, this reflex enables the baby to come out of the prone lying position in preparation for the quadruple position. At the same time, this reflex is gradually integrated by holding the quadruple position and swinging movements (rocking on hands and knees). Children with retained STNR may forgo crawling or do it in a clumsy pattern (hands or feet in unusual positions). In the future, the children with active STNR may have a problem with the fluency of activities requiring shifting the eyes in a vertical line or controlling the sitting posture [5,6]. The onset of TLR is observed prenatally and disappears around the sixth month of life. TLR flexion is caused by the forward flexion of the head with simultaneous flexor tone. TLR extension (in a supine position) is observed as an increase in extensor tone. TLR can entail poor posture, balance, and coordination (careless mistakes and clumsiness).

After entering the preschool period, the child continues intensive motor development, acquiring new motor skills such as jumping, playing ball (catching, throwing, kicking), riding a bike, or balancing. A sign of high neuromotor maturity is increasingly better coordination and control of body movements, including the ability to remain seated for extended periods and concentrate (a skill needed for school education) [5]. Persistent reflex activity can influence the quality of a child’s motor skills, especially coordination and balance. They may be linked to learning difficulties and behavioral problems [9,10]. Incomplete integration of primitive reflexes may cause an involuntary motor response in a reflex pattern. The reflex tensions and uncontrolled movements require conscious and increased control over the child’s motor activities. Their activeness prevents the child from moving fluidly. Non-integrated primary reflexes are described in the literature among the factors that contribute to disorders in the harmonious motor development of the child—they are manifested by clumsiness and frequent learning difficulties [10,11,12,13,14,15,16]. The degree of expression, number, and frequency of reflex activity in healthy children (without a diagnosis of disease or disability) is an indicator of neuromotor maturity.

In a pathological situation, when primitive reflexes are observed in persons with a damaged nervous system (in children with cerebral palsy or after a stroke), strongly expressed automatic reflex movements are still present during motor activities [2,3,4]. The delay of integrating these reflexes at a typical developmental correlates with delays or deficiencies in reaching milestones and a reduced ability to process sensory information [1,12,15]. However, as studies show, a trace form of APR in the population of healthy children is common. According to research, over 90% of children aged 4–6 have at least one of the resisted reflexes present to some extent [1,5,6,12]. Gieysztor [17] showed statistically significant differences in the level of reflex integration between preschoolers (children aged 4–6) and schoolchildren (children aged 7–8). Preschool children are more likely to encounter moderate and higher primary reflex activity. A much larger percentage of children have active reflexes to a low or moderate degree in the school-age group. Grzywniak’s [18] research shows that in school-age children, primary reflexes occur in 55% of the healthy population only at a low or moderate level. Earlier research by Goddard-Blythe [19] shows a similar result—48% of school-age children had some APRs. It should be assumed that in some children, the complete integration of reflexes as the child grows up is a spontaneous process. However, active primary reflexes in a large proportion of school-age children indicate that the process of self-integration of reflexes does not occur at the same rate in all children. Even partial reflexes beyond the period of their physiological occurrence may interfere with normal neuromotor development.

This article presents the influence of active reflexes on the motor skills of preschool children.

The fundamental research question is whether the level of reflex activity correlates with the level of motor performance in terms of balance and coordination skills.

The following hypothesis is tested: an increased index of primary reflexes is associated with decreased motor performance.

An additional research question is whether age and gender are important for a higher index of the integration of primary reflexes and motor skills.

## 2. Materials and Methods

The study is part of the PRACS (Primitive Reflexes and All Children Sphere) project. The project investigates primitive reflexes and their impact on the motor, sensory, and cognitive development in preschool and school-age children. The first article under the PRACS project demonstrates the effect of one of the primitive tonic reflexes (ATNR) on the symmetry of a child’s gait [7]. The second article concerns a child’s perception of their development and difficulties in relation to an adult’s assessment of those difficulties [20]. Finally, the third article in the series presents how the activity of primary reflexes in preschool children correlates with various sensory problems [12]. The research took place in 2016–2019 in three different kindergartens in Lower Silesia. It was carried out by two authors of this article. Both researchers completed special courses entitling them to examine children using select research methods. The reflex test was carried out independently of the MOT test to minimize the risk of non-conscious bias and prevent the children from getting tired during the examination.

The project was approved by the Medical University’s Ethics Committee (approval number KB-626/2018). The parents of all children were informed about the purpose and process of the examination. They were also asked to give written consent for their child to participate in the study and completed information questionnaires about the children.

### 2.1. Participants

The data were collected from 112 children (63 girls and 49 boys) aged 4 to 6 from Lower Silesia preschools (Wrocław, Poland). The mean age of the group was 4.65 (±0.88) years. The characteristics of the participants are shown in Table 1 (61 four-year olds, 34 five-year olds, and 17 six-year olds). The exclusion criteria included confirmation of neuromotor disabilities (cerebral palsy or autism, for example) or pedagogical diagnosis of special needs (intellectual disability, for example).

Each child was assessed with the Primitive Reflex Test [21] (ATNR, STNR, TLR) and the Motor Proficiency Test for children aged 4–6 years (MOT 4–6) [22].

### 2.2. Assessment of Reflex Activity

Tests used to assess the prevalence of primitive reflexes in children were included in the “Test Set for Children Aged 4–7” developer by Sally Goddard-Blythe (INPP, Chester, UK) [21]. The research methodology was described in articles by Goddard [5], Gieysztor et al. [6], Konicarova [14], and Hazza [23]. Due to the specifics of the research, the method of recording the results of the reflex tests sometimes varies among some researchers, but the basis is a set of tests developed by Sally Goddard. The researchers have completed a course enabling them to diagnose active primary reflexes in preschool and early school children. The two-day course conducted following the guidelines of the INPP makes it possible to test children aged 4–7 for active primary reflexes. It also allows conducting exercises aimed at integrating reflexes and improving motor functions. However, it does not authorize individual therapy for children with strongly and maximally expressed reflexes. This examination cannot be used to diagnose diseases or pathologies of the nervous system. This test is intended for children with an undamaged central nervous system structure.

The asymmetrical tonic neck reflex (ATNR) and the symmetrical tonic neck reflex (STNR) were tested in a four four-point kneeling position. In the starting position, shoulders, hips, and knees were flexed to 90°, elbows extended, hands flat, fingers extended, with the head in a neutral position. Next, the tonic labyrinthine reflex (TLR) was tested in a standing position, feet parallel to the width of the hips and hands along the trunk. Before the reflex examination, children were asked to close their eyes.

In the ATNR test, the examiner gently rotated the head passively to the right side and held it for 5 s. Next, the head was slowly rotated back to the midline, and the rotation was repeated on the left side. The whole sequence was performed three times. During the study, it was observed whether the rotation of the head generated changes in the position of the upper limbs (elbow or shoulder bending or hand movements), trunk, and pelvis. The ATNR for both variants: the left (L) and right (R), is presented in Figure 1 and Figure 2.

The STNR was tested for flexion (STNR FLX) and extension (STNR EXT). For STNR FLX (Figure 3), the children were observed for elbows bent and/or pelvis lifted after flexing the child’s head by the examiner. Sometimes children try to extend the knees. For STNR EXT (Figure 4), the children were observed for elbow extending or sitting on the heels after head extending. The trunk movements are also symptoms of active STNR.

In the TLR test, the children were asked to flex and extend the head by themselves. For the TLR FLX (Figure 5) test, the children were asked to bend the head slowly, as if “looking at their feet,’’ and stand in this position for 10 s. Next, the children were asked to tilt the head back as far as possible for 10 s (TLR EXT test, presented in Figure 6). Compensation, such as fist-clenching, upper limb movements, knee deflection (in TLR FLX), or climbing on toes (in TLR EXT), and/or general balance disturbance, constituted symptoms of an active TLR.

No reaction (no body movements) to the change in the head position is a symptom of the complete integration of APRs in the child’s central nervous system. The level of the reflex activity is determined based on the quantitative and qualitative compensatory movements that occur in response to the head movement. Emotional reactions (screaming or confusion), changes in the rhythm of breathing, frowning, or pursed lips may also indicate that the reflex is still active. The reflexes were assessed on a five-step scale (0—lack of reflex, 1—low activity, 2—medium activity, 3—high activity, 4—maximum activity). The higher the score, the worse reflex integration. The maximum total score in the study is 24 points. The total score of the examination of all reflexes was converted to the level of reflex activity on a scale from 0 to 4, presented in Table 2.

### 2.3. Motor Proficiency Test (MOT) Assessment

The Motor Proficiency Test for children aged 4–6 (MOT 4–6) includes 18 items [22]. The MOT 4–6 is a standardized test developed in a six-month study of 548 German children. The MOT 4–6 refers to a norm. The tool has a high assessment protocol efficiency [24]. The MOT test has been constructed in such a way that some tasks are easier and others more difficult so that it can be used to assess children in the 4–6 age group. All tasks included in the MOT test require maintaining balance, good coordination, and concentration from the children. The tests are repeatable, easy to evaluate and write down in the research protocol, and adequately matched to the skills of preschool children, which best testify to their physical performance (it is not, for example, muscle strength or endurance). These tasks are divided into four performance areas: (a) stability, (b) locomotion, (c) object control, and (d) fine movement skills. Tasks divisions are shown in Table 3.

The tasks were assessed on a three-point rating scale, where 0 means that the task was not completed, 1—the task was completed to about 50% of the child’s capacity, 2—the skill is fully mastered. All task points after adding up give a total score of 34 points. The higher the children scored in the MOT 4–6 assessment, the higher the movement skill level. The points in the MOT test were transferred to a five-point scale describing child development as accelerated, very good, normal, delayed, or altered (Table 4).

The MOT test was deliberately chosen by the authors of this study to test the motor skills of children. Its focus on the study of balance and coordination aspects allows for its results to be compared with the study of reflexes, which also assess the neuromotor maturity as evidenced by these areas of physical performance.

### 2.4. Statistics

The statistical analysis was carried out using Statistica version 13.0. Descriptive statistics were computed for all variables. Arithmetic means and standard deviation were calculated. The distribution was determined by the Shapiro–Wilk test. It was found that the distribution of the data for MOT results was normal. However, the reflex results significantly deviate from the norm. The correlation coefficient between the final MOT result and the total score for reflexes was determined. The relationship between age, reflex level, and motor performance was investigated using multiple regression. Linear regression calculations were performed to assess the influence of age and gender on the MOT 4–6 results (GLM—general linear model). The differences between the reflex scores for boys and girls were tested with the Mann–Whitney U test. The correlation of BMI with physical fitness and the reflex test results was also analyzed. All parameters were considered statistically significantly different if *p* < 0.05.

## 3. Results

### 3.1. Reflex Activity Level in the Examined Group

The total score (max. 24 points) for reflex examination was converted to the level of reflex activity. The results show that 7.1% of the examined children have no retained primitive reflexes. A significant part (58%) of the preschool population has a low level of reflex activity. In 27.7% of children, a medium level of reflex activity was observed. In 5.4% of participants, a high level of reflex activity was found, while the maximum level was demonstrated by 1.8% of subjects. The results of the reflex activity level are shown in Figure 7.

### 3.2. Results of APR Examination

The most frequently occurring reflexes were ATNR L (observed in 80.4% of children) and ATNR R (in 73.2%), and TLR EXT (in 74.1%). The least frequent one was the STNR FLX (observed in 44.6% of children). The reflexes most frequently expressed at the maximum intensity were TLR EXT (17% of children), ATLR L (10.7%), and the least STLR FLX and STLR EXT (2.7%). The percentage results are shown in Table 5

### 3.3. MOT 4–6 Results

The preschoolers’ final results in the MOT 4–6 are shown in Figure 8.

The most difficult tasks for children included carrying balls from one box to another (No. 6), reverse balance (No. 7), and throwing at a target disk (No. 8)—about 80% of children did not complete these tasks. The easiest tasks were standing up while holding a ball on the head (No. 16) and passing through a hoop (No. 10). Task 16 was completed by 91.1% of children, and task 10 by almost 76%. Table 6 and Table 7 present the percentage results of the maximum and minimum scores achieved by children in the MOT 4–6.

### 3.4. The Results of the Motor Proficiency Test and Reflex Activity

Statistical analysis showed an inverse correlation between the number of points in the test of reflexes and motor efficiency (MOT 4–6) at *p* < 0.05 (−0.33). Thus, children with increased reflex activity presented a lower level of motor efficiency.

There is also a visible correlation between motor efficiency (MOT level) and the activity of individual reflexes. A statistically significant correlation is the strongest for STNR EXT and TLR FLX (−0.32 and −0.29). The correlation coefficient between the result of the motor efficiency level and the reflex activity is presented in Table 8.

The correlation was also found for individual MOT 4–6 tasks and reflex activity level. Tasks forward balance (No. 1), reverse balance (No. 7), jumping jacks (No. 13), jumping over a cord (No. 14), and rolling around the long axis of the body (No. 15) correlated most strongly with the total score of tonic reflexes. The correlations of individual tasks from MOT 4-6 with the reflex activity level are shown in Table 9.

### 3.5. Relationship between Age, Reflex Activity and Motor Perfomance

In a multiple regression model, the MOT 4–6 final scores were significantly related to the set of two predictors, i.e., age and APR (F = 32.6, df = 2, *p* < 0.0001, adjusted R2 = 0.36). This analysis showed that both age (positively) and reflex integration (negatively) affected the final score of the MOT test in a statistically significant manner. Furthermore, the effect of age was stronger, as reflected by the beta coefficient. Concluding, it has been shown that the older the child and the lower the reflex activity, the score of the child’s motor skills is higher. The results are shown in Table 10.

### 3.6. Relationship between Age, Gender, and Motor Performance

A GLM analysis was performed to assess the effect of age, gender, and the interaction between the two predictors on the MOT 4–6. The relationship between the motor performance and the set of predictors was highly significant (F = 20.67, df = 3, *p* < 0.0001, adjusted R2 = 0.35). Furthermore, it was established that age had a positive and statistically significant influence on motor performance. On the other hand, both gender and the interaction gender x age had an insignificant effect. The results are shown in Table 11.

### 3.7. Differences between Girls and Boys in the Reflex Activity and MOT 4–6

The Mann–Whitney U test was carried out to determine whether girls and boys differed in motor performance and reflex activity level. The relationship between gender and reflex integration was not found. When the children were divided into age groups, the Mann–Whitney U analysis showed no differences in the MOT 4–6 score (the results of girls and boys did not differ statistically).

### 3.8. The Relation of Results of the Reflex Level and MOT Test to BMI

Based on the measurements of height, weight and the age of children, the body mass index (BMI) was calculated. No statistically significant relationships were found after analyzing the correlation between BMI for children’s motor skills and reflex activity.

## 4. Discussion

The aim of our study was to verify if the level APRs are related to motor performance in terms of balance and coordination skills. The study of APR in relation to motor skills is a kind of innovative and unique approach to the subject. Previous studies have mostly focused on showing the relationship between unquenched reflexes and problems with reading, learning, and difficulties in maintaining concentration. We hypothesized that the increased index of primary reflexes is associated with decreased motor performance. The results were also analyzed in terms of age and gender. This paper also aims to show that the child’s motor performance in terms of good balance and coordination skills are a fundamental element for the proper pace of neuromotor maturation and the correct development of many areas of the child’s functioning (such as learning to read, focus, and speech development).

### 4.1. The APRs Releted to Motor Skills

The correlation between the activity of tonic reflexes and the final score of the MOT 4–6 shows an inverse relationship between the degree of non-integration of reflexes and motor efficiency of preschool children. The same fact is stated in research by Gieysztor [6], who also examined preschool children (a group of 35 children) for the presence of tonic reflexes and motor performance using the MOT 4–6. Our new research was carried out on a much larger group of children (112 study participants) than most of the current reflex studies. We also extended the analysis of the results to establish the correlation for all three reflexes with the MOT level and for individual movement tests with the level of active reflexes. Most of the reflexes significantly correlated with the level of motor performance of the children. The MOT test trials, which correlated most strongly with the final score for reflex activity, mainly assessed the skills in group B, i.e., locomotor (dynamic) skills. In tests such as balancing forward and backward, jumping over a cord, and jumping jacks, balance and coordination are essential skills needed to complete the task correctly. Active tonic reflexes mean that the head movement may cause uncontrolled tension and/or movement of the lower and upper limbs and the trunk, thus preventing the child from effectively carrying out a motor task requiring total balance and body control.

So far, only a few studies show that the APRs co-exist with disturbances in child relationship of active reflexes with motor and posture [6,7,12,25]. The confirmation of this finding is described by Gieysztor et al. [7] in an article on gait in preschool children. ATNR, which was active in them, affects the symmetry of the pelvic alignment during gait and thus its quality. In our research, we also observed an inverse correlation between the level of APRs and the results obtained in the forward and backward balancing test. In studies involving school children with reading difficulties, Bilibaj [10] notices a relationship between ATNR activity and difficulties in eye–hand coordination. Taylor et al. [16], who studied the relationship of persistent tonic reflexes and the Moro reflex with the achievement, report that the APRs were linked with carelessness and difficulty sitting still. In studies using the sensory profile, Pecuch et al. [12] show that children whose parents notice sensory problems such as dyspraxia and postural disorders have an increased degree of active primary reflexes. Parents of children tested for the activity level of six selected reflexes, including those described in this article, were asked to complete the sheet describing the child’s sensory profile. Based on the research, it was established that there is an inverse relationship between the activity of reflexes and the parents’ noticing problems with coordination and balance in their children, the ability to assess the distance, and the sense of the body in space. They also noticed difficulties such as frequent stumbling and rapid fatigue. Hazzaa [23] also described the problem of an increased number of reflexes and poor balance in children with learning difficulties (manifested by problems with maintaining concentration, poor memory, reading, and writing). Children from the study group performed significantly worse in terms of the one-leg stand test results than the group of children with no learning difficulties. Alibakhshi’s research [26] shows that in school-age children with specific learning difficulties, these problems were also accompanied by poor fine motor skills and the persistence of ATLR statistically more often than in children without learning difficulties. Matuszkiewicz’s research [27] shows that active primitive reflexes can also cause speech development problems. Research also indicates many other developmental aspects, including the formation of scoliosis [8,25]. While examining girls with ADHD, Koniciarova [14] observed a correlation between ATNR with anxiety, impulsive behavior, and perfectionism, present in their parents’ opinion. In these studies, girls were assessed with the CPQ test, which indicated disorders typical of ADHD. The final score of this test and learning problems correlated with active STNR. In Hickey [1] research, the active ATNR L has a significant relationship within the Opposition/Defiance (the inattention subtest of the SNAP-IV). Accardo [28] found a link between characteristic toe walking and active TLR in individuals on the autism spectrum. Research shows that prolonged and increased occurrence of primary reflexes past infancy may be an alarming signal of future development problems or various types of disorders [3]. Moreover, in high-risk newborns, the occurrence of primary reflexes differs qualitatively and quantitatively from non-risky children [29]. Gieysztor [6] showed that in children born before term (preterm infants), integration of primary reflexes and physical performance is lower compared to children born at term. The persistent occurrence of primary reflexes can be observed in both children and adults [30]. It means that their occurrence may persist throughout life and thus affect its quality. Demiy et al. [20] show that children with motor difficulties concerning balance, coordination, or concentration find difficulties in these areas of motor skills in themselves. They even notice them much more than their parents. Usually, however, the child’s “clumsiness” or emotional difficulties and problems with developing skills necessary for effective participation in the educational process, such as writing and understanding the read text, are unknown.

### 4.2. APRs in Preschool Children in Relation to Age

The presence of primary reflexes is described in the literature mainly in children with cerebral palsy [2,4]. In recent years, there have been several studies on primary reflexes in children in the healthy population and children with specific developmental difficulties [1,5,6,12]. Research shows that primary reflexes in a trace form concern a significant percentage of the population of children in preschool and school-age. In our research, APRs were found in 92.9% of the examined children. Hickey and Feldhacker [1] show that 100% of children aged 4–6 had at least one retained reflex. The same finding is reported by Goddard, who studied children aged 4–5 [5]. Pecuch et al. [12] examined six APRs (ATNR, STNR, TLR, Moro reflex, and skin reflexes: Palmar and Galant) in children aged 4–6, and the study shows at least one of these reflexes in 98% of preschool children. In Gieysztor’s research [17], at least one of the three active tonic reflexes was detected in 89% of preschoolers (age 4–6), but in 65%, they had a barely residual degree. In 25% of children, the reflexes demonstrated at a higher level (3–4). In the Goddard study, where the study group consisted of children aged 4–5 moderate to maximum reflexes occurred in even more children (approximately 70–80% of the sample depending on the reflex). In our research, 58% of children have a low level of reflex activity. The differences in the results may be due to the fact that the Goddard study group does not take into account the six-year olds. According to Hickey [1], the most popular reflex in the preschool population was STNR, as it was found in 81.4% of children. Gieysztor [6] found that ATLR L was least integrated with age (this reflex was present in 78% of preschool children and 34% of schoolchildren). The best-integrated reflex was TLR FLX (full integration reached by 65% of preschoolers and 95% of schoolchildren). Our research shows that the STNR FLX is fully integrated in 55.4% of preschool children (together with TLR FLX, it is the best-integrated reflex). ATLR L occurred in 81.7% of children, and STNR EXT was present in 67.9% of children (the result is similar to that obtained in Hickey’s research). 

Therefore, the question arises whether primary reflexes always constitute a severe problem to the psychomotor development and when it should become a reason for increased observation of the child and possibly therapy.

In our research, the results of multiple regression show that there is a tendency to achieve better motor performance in older children with a simultaneous decrease in the level of reflex activity. Studies comparing preschool and schoolchildren show that the level of reflex activity decreases with age [17]. There is a lack of long-term studies demonstrating unambiguously at what age children completely integrate their reflexes. However, it could be assumed that slightly expressed reflexes in preschool children may integrate completely by the school age. However, Grzywniak [31] reports that in children with learning difficulties, reflex activity increases with age (and there is no spontaneous integration). In response to the above question posed by the researchers, it is necessary to state that special attention should be paid to children with increased activity of primary reflexes and specific sensorimotor, emotional, or educational difficulties. Based on own research, it can also be concluded that if the child’s reflexes do not integrate automatically with age, their motor skills will not develop properly.

### 4.3. APRs Integrating Therapies

It is worth noting that researchers dealing with primary reflexes have developed strategies and exercises that integrate reflexes. The best-known strategies and techniques proposed by Swietlane Masgutowa [32] and Sally Goddard-Blythe [5,33] have been researched so far. Few studies describe interventions in the form of exercise programs or techniques to extinguish active primal reflexes. Goddard’s research has shown that appropriate movement exercises positively affect the integration of primary reflexes in preschool and school-age children [5]. The exercises are programmed to allow the child to repeatedly experience movement patterns as opposed to those generated by non-integrated reflexes. These exercises improve balance and coordination skills and proprioception (a sense of their own body in space). Grzywniak reported the same finding in her research [34]. Grigg et al. [35] describe the positive effects of using the Rhythmic Movement Training by parents to integrate primary reflexes in children aged 7–12. Melillo et al. [36], in a study of ADHD patients, observed a significant improvement in reflex reduction and improvement in motor and cognitive functions after using a 12-week training program. In a study by Wagh et al. [37], children aged 12–24 m with confirmed CP also exhibited strongly expressed primitive reflexes. Researchers showed that a specific reflex-quenching program lasting six weeks in such children also resulted in improvement, as demonstrated by the HINE test (clinical neurological examination to assess the motor development of infants). In all interventions, after using techniques integrating the primary reflexes, improvements in reflex retesting and motor function tested using various tests were noted.

### 4.4. Motor Skills of Preschool Children

Our research shows that the most difficult test to perform from the MOT 4–6 set of tasks was carrying the balls from one box to another (No. 6)—87.5% failed the test and reverse balance (No. 7)—85.7% of children did not receive any point for this task. The easiest task was to stand up and sit down with a ball (No. 16) and passing through a hoop (No. 10). Gieysztor’s research [6] confirms that these two tests were the most difficult to perform for children in the preschool group, and task No. 16 was the easiest. Cools et al. [24] examined the distribution of test results in the MOT test in the group of preschool children. In our study, the children’s MOT 4–6 mean performance was 16.7 (SD = 5.8). In Cools et al., mean performance was 19 (SD = 4.8), in Gieysztor—15 (SD = 4.7). Cool et al. and Gieysztor’s studies on the distribution of children’s performance were comparable to our research findings. Most children were in the “normal development” range (in Cools et al.—75%, Gieysztor—59%, our research—60%). In our research, 6.3% of children were classified under “altered development” (in Cools et al.—4%, Gieysztor—9%). In our research, “very good development” was found in over 6%; in Cools et al.—1%; and Gieysztor—3%. In Cools et al. and Gieysztor’s research, no child obtained an “accelerated development” result. In our study, this result was found in 1.8% of cases. There is a tendency for the number of children with the normal result to decrease toward the extreme ones. Perhaps it is the effect of the increasingly limited spontaneous activity of children toward a stationary way of spending free time. It also appears that in recent years the availability of additional sports activities, where even very young children can significantly improve their motor skills, has increased. Reference to motor performance could be expanded in a study considering the analysis of other correlates such as time spent at home in front of a computer/smartphone and participation in extra-curricular activities focused on physical activity. However, this is an issue that should be addressed in a separate article, due to the fact that the context of the issue is too broad. Preedy’s [38] research on the motor skills of preschool children showed that poor physical development impacts readiness for school, learning achievement, behavior, and social development.

### 4.5. APRs and Motor Skills Releted to Gender and BMI

The GLM test shows that gender and the interaction gender and age had an insignificant effect on motor performance. The Mann–Whitney U analysis showed no relationship between gender and the level of reflex integration. Hickey’s [1] research shows that males demonstrated more reflex activity than females in most examined reflexes. Pecuch [12] compared the data between girls and boys. In the total score of the reflex examination, girls showed a higher degree of integration. In Gieysztor’s [6,17] research, girls also obtained scores indicating a higher level of reflex integration and better motor skills. According to the authors of the MOT test based on population studies, there is no need for a separate norm in the assessment of motor performance for boys and girls in preschool-age [39]. Due to the lack of significant differences between the sexes in total motor activity, these tests are used for boys and girls on the same scale. However, based on the research, there is a tendency to obtain a higher index of reflexes integration and motor fitness by girls.

The index of physical fitness and the level of reflexes did not correlate significantly with the BMI of children. Moreover, Gieysztor [6], who tested the level of reflexes and motor performance of children with the MOT test, did not report such dependencies. It seems reasonable because parameters such as BMI appear to have a greater impact on aspects of physical fitness such as strength or speed, not parameters related to balance and coordination.

## 5. Conclusions

A relationship between APRs and motor skills shows that the presence of active primary reflexes in preschool children can and should be an indicator of neuromotor development. APRs are expected to become integrated with age parallel to motor skills development. It has been shown that the older the child and the lower reflex activity, the score of the child’s motor skills is higher. However, when the process of spontaneously integrating reflexes is not run properly, it may disturb other areas of the child’s functioning in terms of acquiring motor skills (related to balance and coordination), education, and social life.

An important piece of information for persons who work with children is the fact that there are methods of therapeutic work on the integration of reflexes. Screening for APRs among children may be helpful in selecting appropriate integrating exercises at preschool or during individual therapy. It provides a chance to reduce the activity of reflexes and thus improve the balance and coordination of children resulting in better use of their educational potential and the ability to self-regulate emotional processes and social behavior. Evidence-based research to address the problem of primary reflexes in children should be the focus of research for scientists in various fields.

## Figures and Tables

**Figure 1 brainsci-11-00967-f001:**
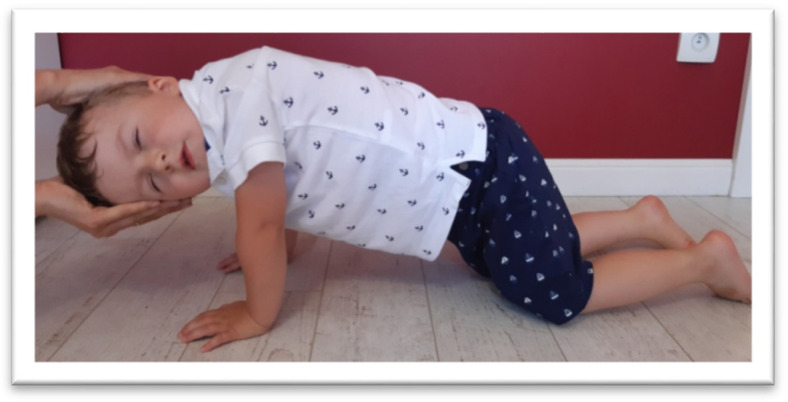
The examination of ATNR L. On the facial side, there is an excessive extension of the upper limb.

**Figure 2 brainsci-11-00967-f002:**
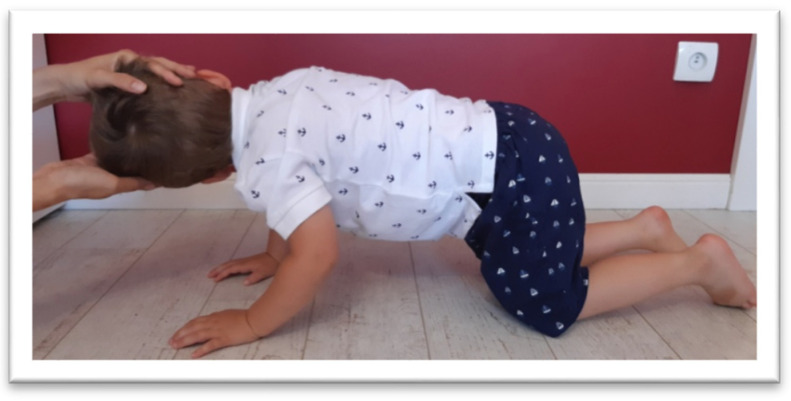
The examination of ATNR R. Upper limb flexion is visible on the occipital side, and pelvic movement/descent may occur.

**Figure 3 brainsci-11-00967-f003:**
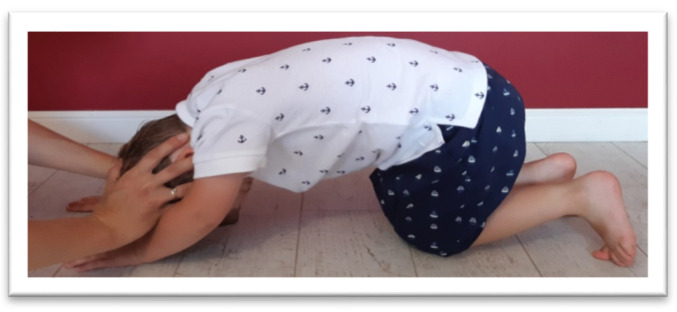
The examination of STNR FLX. The flexion of the head is accompanied by the flexion of the arms, and hip extension may appear.

**Figure 4 brainsci-11-00967-f004:**
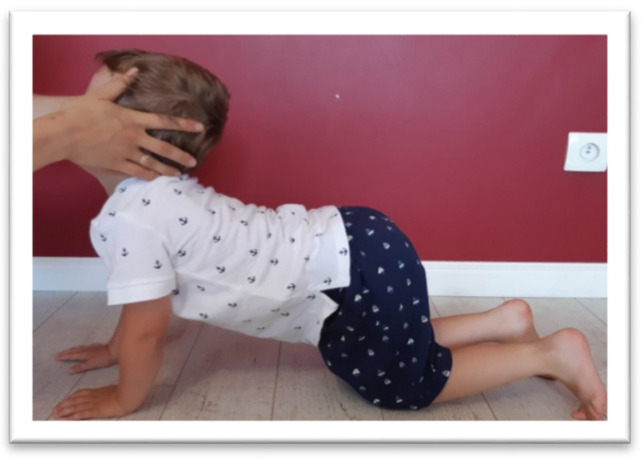
The examination of STNR EXT. The straightening of the head may accompanied by sitting on the heels and arm strength.

**Figure 5 brainsci-11-00967-f005:**
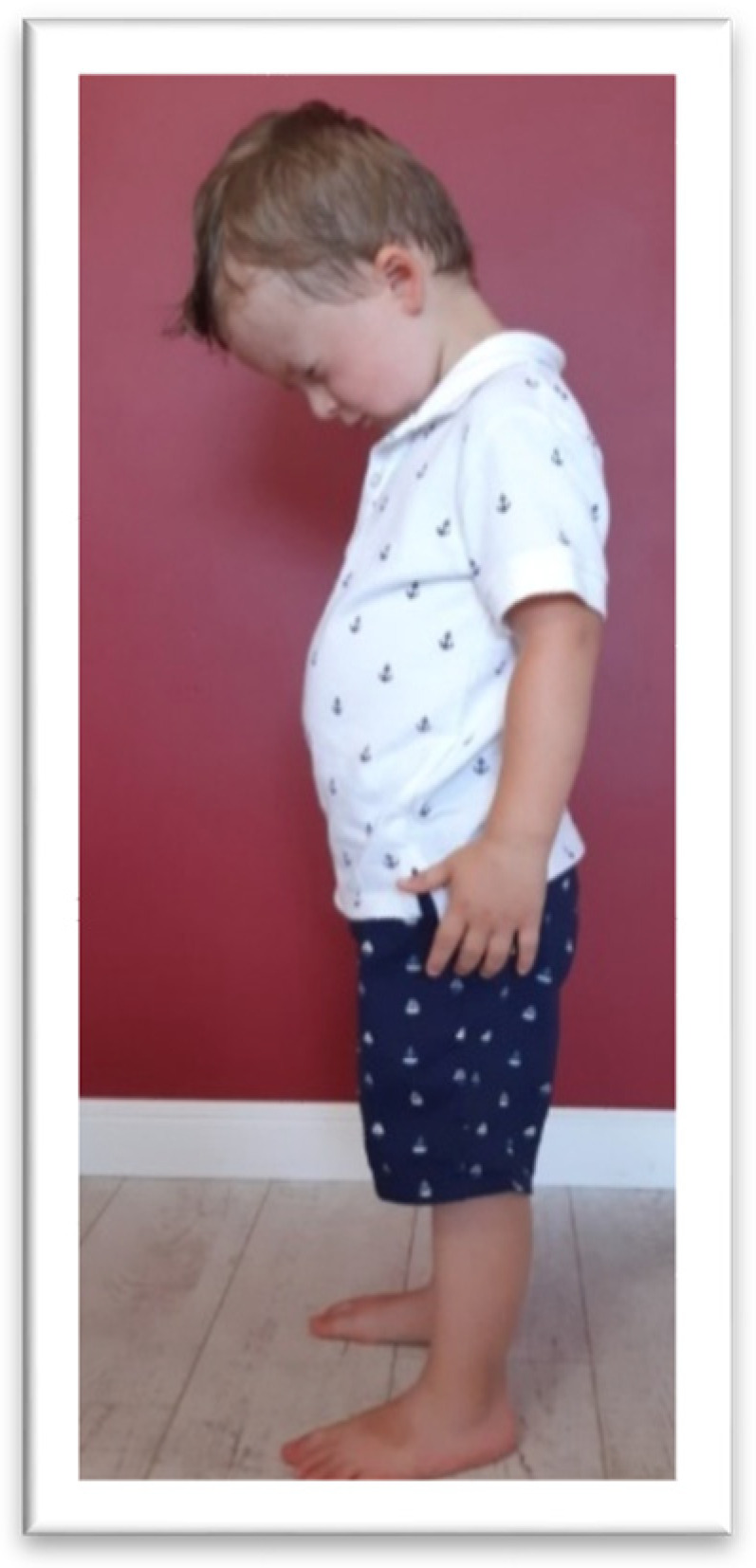
The examination of TLR FLX. The bend of the head is accompanied by a bent knee and slumped posture.

**Figure 6 brainsci-11-00967-f006:**
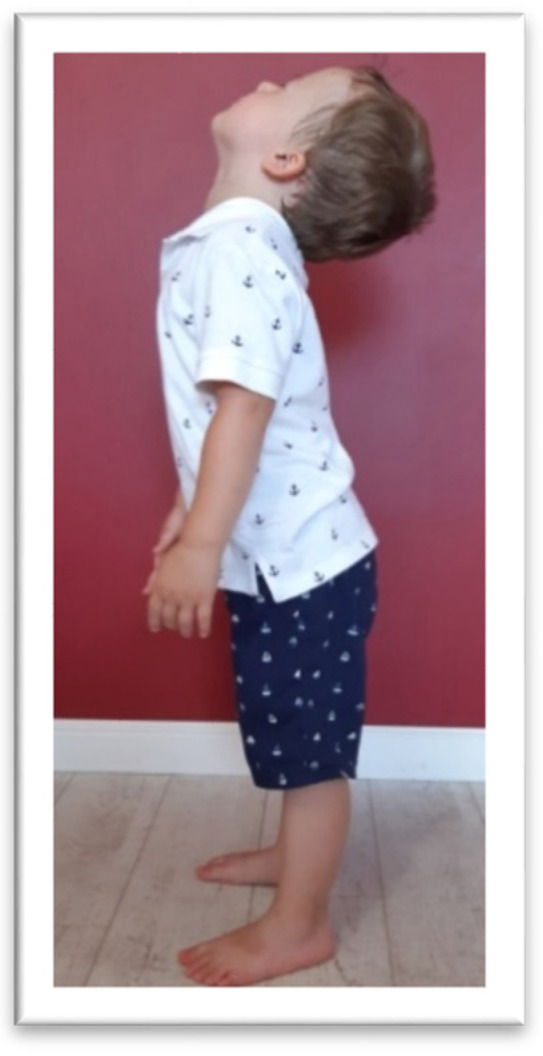
The examination of TLR EXT. The tilting back of the head is accompanied by climbing on the fingers and folding the hands back.

**Figure 7 brainsci-11-00967-f007:**
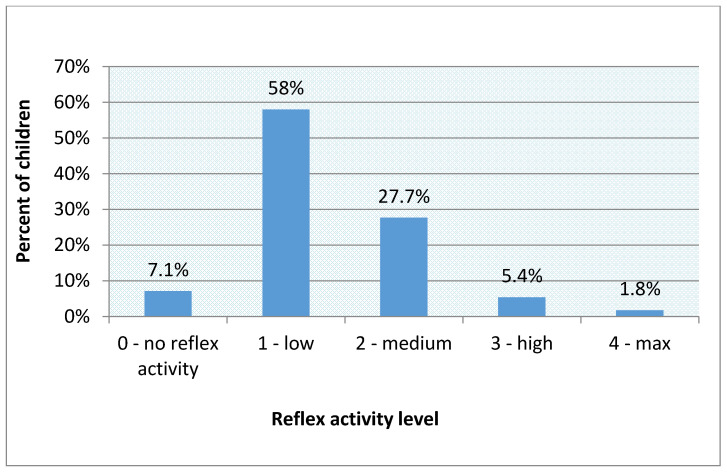
The results concerning the level of reflex activity.

**Figure 8 brainsci-11-00967-f008:**
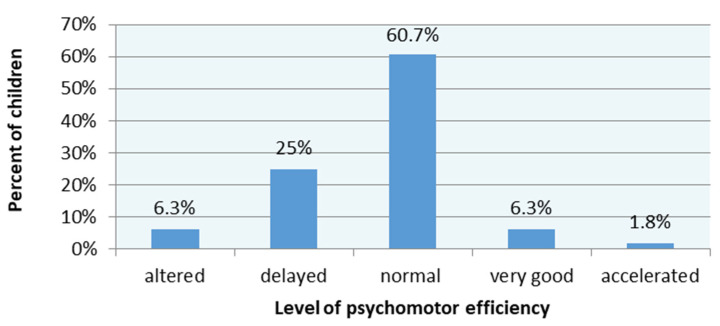
The MOT 4–6 final results.

**Table 1 brainsci-11-00967-t001:** Characteristics of the participants.

Parameter	4 Years	5 Years	6 Years
Age, mean ± SD Girls	4.1 ± 0.2 38	5.1 ± 0.2 15	6.24 ± 0.4 10
Boys	23	19	7
Height, mean ± SD [m]	103.7 ± 6.1	111.1 ± 5.5	118.69 ± 5.9
Weight, mean ± SD [kg]	17.6 ± 3.8	19.4 ± 3.1	22.92 ± 4.7
BMI, mean ± SD [kg/m²]	18.3 ± 13.6	15.8 ± 2.2	16.19 ± 3.1

**Table 2 brainsci-11-00967-t002:** The degree of primitive reflex integration scale.

Final Score in APR Examination	Reflex Activity Level
0–1	0—no activity
2–8	1—low activity
9–14	2—medium activity
15–21	3—high activity
21–24	4—maximum activity

**Table 3 brainsci-11-00967-t003:** MOT 4–6 items.

Description of Items	
0. Forward jump in a hoop * 1. Forward balance (b) 2. Placing dots on a sheet (d) 3. Grasping a tissue with toes (d) 4. Sideward jump (b) 5. Catching a stick (c) 6. Carrying balls from one box to another (b,c) 7. Reverse balance (b) 8. Throwing at a target disk (c) 9. Collecting matches (d)	10. Passing through a hoop (a,b) 11. Jumping in a hoop on 1 foot, standing on 1 leg (a,b) 12. Catching a tennis ring (c) 13. Jumping jacks (b) 14. Jumping over the cord (b) 15. Rolling around the long axis of the body(a,b) 16. Standing up while holding a ball on the head (b,c) 17. Jumping and turning in a hoop (a,b)

* The first item was not rated because it was used to accustom the child to the test situation. (a) stability; (b) locomotion; (c) object control; (d) fine movement skills.

**Table 4 brainsci-11-00967-t004:** The MOT test range scale.

Final Score in MOT Test	Level of Psychomotor Efficiency
0–8	0—altered
9–15	1—delated
16–25	2—normal
26–29	3—very good
30–34	4—accelerated

**Table 5 brainsci-11-00967-t005:** The results of the examination of reflexes.

SCALE	ATNR R	ATNR L	STNR FLX	STNR EXT	TLR FLX	TLR EXT
0	26.8%	19.6%	55.4%	32.1%	44.6%	25.9%
1	28.6%	41.1%	28.6%	32.1%	33.0%	28.5%
2	23.2%	19.7%	6.2%	19.6%	12.5%	18.8%
3	13.4%	8.9%	7.1%	12.5%	4.5%	9.8%
4	8.0%	10.7%	2.7%	2.7%	5.4%	17.0%
MEAN	1.5	1.5	0.7	1.2	0.9	1.6
SD	1.2	1.2	1.0	1.1	1.1	1.4

ATNR R/L—asymmetrical tonic neck reflex (right/left); STNR FLX/EXT—asymmetrical tonic neck reflex (flexion/extension); TLR FLX/EXT—tonic labyrinthine reflex (flexion/extension).

**Table 6 brainsci-11-00967-t006:** The results of MOT 4–6 for tasks 1–8.

				TASK				
SCALE	1	2	3	4	5	6	7	8
0	33.0%	18.2%	31.5%	49.1%	28.6%	87.5%	85.7%	74.1%
1	35.7%	23.5%	18.2%	34.8%	58.9%	10.7%	10.7%	23.2%
2	31.3%	58.3%	50.3%	16.1%	12.5%	1.8%	3.6%	2.7%
MEAN	1.0	1.4	1.2	0.7	0.8	0.2	0.2	0.3
SD	0.8	0.8	0.9	0.7	0.6	0.4	0.5	0.5

**Table 7 brainsci-11-00967-t007:** The results of MOT 4–6 for tasks 9–17.

				TASK					
SCALE	9	10	11	12	13	14	15	16	17
0	57.1%	7.1%	42.0%	45.2%	26.8%	23.2%	11.6%	0.9%	23.5%
1	28.6%	17.0%	26.8%	27.4%	33.9%	33.0%	38.4%	8.0%	29.8%
2	14.3%	75.9%	31.2%	27.4%	39.3%	43.8%	50.0%	91.1%	46.7%
MEAN	0.6	1.7	0.9	0.8	1.2	1.2	1.4	1.9	1.3
SD	0.7	0.6	0.9	0.8	0.8	0.8	0.7	0.3	0.9

**Table 8 brainsci-11-00967-t008:** The correlation coefficient between reflex activity and MOT level.

Reflex	Correlation Coefficient
ATNR R	−0.204 *
ATNR L	−0.161
STNR FLX	−0.201 *
STNR EXT	−0.317 *
TLR FLX	−0.294 *
TLR EXT	−0.157
TOTAL SCORE	−0.327 *
REFLEX ACTIVITY LEVEL	−0.306 *

* *p* < 0.05; ATNR R/L—asymmetrical tonic neck reflex (right/left); STNR FLX/EXT—asymmetrical tonic neck reflex (flexion/extension); TLR FLX/EXT—tonic labyrinthine reflex (flexion/extension).

**Table 9 brainsci-11-00967-t009:** The correlation coefficient of reflex activity level with individual tasks from the MOT test.

MOT 4–6 Task	Correlation Coefficient
1. Forward balance	−0.215 *
2. Placing dots on a sheet	−0.125
3. Grasping a tissue with toes	−0.190
4. Sideward jump	−0.010
5. Catching a stick	−0.072
6. Carrying balls from one box to another	−0.100
7. Reverse balance	−0.256 *
8. Throwing at a target disk	−0.092
9. Collecting matches	−0.055
10. Passing through a hoop	−0.028
11. Jumping in a hoop on 1 foot, standing on 1 leg	−0.030
12. Catching a tennis ring	−0.072
13. Jumping jacks	−0.262 *
14. Jumping over a cord	−0.288 *
15. Rolling around the long axis of the body	−0.251 *
16. Standing up while holding a ball on the head	−0.158
17. Jumping and turning in a hoop	−0.090

* *p* < 0.05.

**Table 10 brainsci-11-00967-t010:** The results of the multiple regression model of the effect of age and APR (both Box–Cox transformed) on MOT 4–6.

	Coefficient	95% CI	Beta (ß)	95% CI	*t*	*p*
**Intercept**	−50.67	(−69.15; −32.19)			−5.43	0.000
**Age**	91.86	(67.81; 115.91)	0.57	(0.42; 0.73)	7.57	0.000
**APR**	−0.66	(−1.23; −0.09)	−0.17	(−0.33; −0.02)	−2.30	0.023

**Table 11 brainsci-11-00967-t011:** The results of the GLM analysis of the effect of age and gender on MOT 4–6.

	Coefficient	95% CI	Beta (ß)	95% CI	*t*	*p*
**Intercept**	−58.69	(−77.70;−39.69)			−6.12	0.000
**Gender (=1)**	−15.02	(−34.02; 3.98)	−2.55	(−5.78; 0.68)	−1.57	0.120
**Age**	99.28	(74.28; 124.28)	0.62	(0.46; 0.78)	7.87	0.000
**Gender (=1) × Age**	19.13	(−5.87; 44.13)	2.47	(−0.76; 5.69)	1.52	0.132

## Data Availability

The data supporting reported results can be obtained directly from the correspondent author who deals with data storage.

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
