# Peer review of "Primitive Reflex Activity in Relation to Motor Skills in Healthy Preschool Children"

_brainsci, 2021, doi:10.3390/brainsci11080967_

Round 1

Reviewer 1 Report

Dear authors

This is a very interesting piece of work. I trust my comments will assist you with strengthening the manuscript. 

  1. The introduction needs more information about the relationship between the reflexes and motor skills. Have other studies examined this relationship in other populations? As it stands, the introduction does not clearly indicate the importance or reason for this study.
  2. In line 73, the children/participants are referred to as ‘subjects’ – this should be changed.
  3. Is it possible to provide some examples of the motor disabilities that resulted in exclusion? Given the nature of this study, this information is helpful.
  4. Table 1 is redundant as this information is already provided in the paragraph prior. I recommend removing.
  5. Is it possible to provide more details about the course that the researchers completed to be skilled in testing the children’s reflexes? This, I would imagine, is a highly specialised skill that is in other circumstances, reserved for medical practitioners.
  6. There appears to be no references listed for the testing of reflexes. The authors have done well to add figures and explain the test, but it would be good to note whether these tests are used in practice or clinical settings. On this point, the Figure captions do need to be more descriptive to be helpful to the reader.
  7. Table 3 is helpful but a description of what the letters in brackets mean would be helpful to add as footnotes.
  8. It appears that the children’s height and/or weight were not measured. This is an important consideration, particularly with motor skills as some skills tend to vary according to weight status. Was this the case for the sample?
  9. The text reporting the MOT results is quite vague – it would be helpful to the reader to explicitly mention the specific skills instead of referring to them by numbers from the table.
  10. It is expected that the older children would exhibit greater motor skill proficiency – the analysis in section 3.5 is likely not necessary.
  11. As it stands, there are two sections numbered 3.6.
  12. The discussion section needs to draw more attention to what the results mean and what the implications are for this study – there is substantial repetition of results and no discussion at all about other correlates that are important to consider when assessing motor skills. For example, other factors that are associated with motor development include overweight/obesity, motor skill teaching, access to facilities etc. These factors have not been acknowledged and may explain some of the findings.

Author Response

Dear Reviewer
In the attachment I am sending the replies to your comments. Thank you very much for them because they have definitely contributed to the improvement of the quality of the manuscript.

  1. The introduction needs more information about the relationship between the reflexes and motor skills. Have other studies examined this relationship in other populations? As it stands, the introduction does not clearly indicate the importance or reason for this study.

Answer: After reading your comments and other reviewers, I also noticed that the introduction was too minimalistic by me. Currently, I have enriched it significantly with a description of the mechanisms that characterize individual reflexes and what they influence. I hope that now the connections between the occurrence of reflexes and the disruptions of specific skills are much more visible and understandable. I also partially moved the paragraph that talked about the occurrence of reflexes in the population of children, and what their occurrence, according to the authors of various studies, may influence. In the previous version, this paragraph was part of the discussion.

  1. In lin3, the children/participants are referred to as ‘subjects’ – this should be changed.

Answer: Words have been replaced by participants.

  1. Is it possible to provide some examples of the motor disabilities that resulted in exclusion? Given the nature of this study, this information is helpful.

Answer: An example of a disabilities is included in the text now.

  1. Table 1 is redundant as this information is already provided in the paragraph prior. I recommend removing.

Answer: The table has not been removed because the reviewer 3 suggested that it should be enriched with data on weight, height and BMI.

  1. Is it possible to provide more details about the course that the researchers completed to be skilled in testing the children’s reflexes? This, I would imagine, is a highly specialised skill that is in other circumstances, reserved for medical practitioners.

Answer: It is a two-day training conducted by persons authorized to conduct training by the INPP institute. This course gives the opportunity to test children for the presence of active primary reflexes and to conduct group exercises aimed at integrating reflexes and improving motor functions. However, it does not entitle to individual therapy for children with strongly and maximally expressed reflexes. This examination is intended for doctors, physiotherapists, educators and people working with children on a daily basis. More information on: https://www.inpp.org.uk/

  1. There appears to be no references listed for the testing of reflexes. The authors have done well to add figures and explain the test, but it would be good to note whether these tests are used in practice or clinical settings. On this point, the Figure captions do need to be more descriptive to be helpful to the reader.

Answer: References have been completed. These tests can be used both in clinical and practice. The descriptions for the photos have also been expanded.

  1. Table 3 is helpful but a description of what the letters in brackets mean would be helpful to add as footnotes.

Answer: Letter designations have been added in the footnote below the table.

  1. It apears that the children’s height and/or weight were not measured. This is an important consideration, particularly with motor skills as some skills tend to vary according to weight status. Was this the case for the sample?

Weight, height and BMI were measured, however, these data, in statistical analysis, did not correlate with measurable parameters. This seems understandable and also explained in the discussion.

  1. The text reporting the MOT results is quite vague – it would be helpful to the reader to explicitly mention the specific skills instead of referring to them by numbers from the table.

Answer: Numerals have been replaced with words

  1. It is expected that the older children would exhibit greater motor skill proficiency – the analysis in section 3.5 is likely not necessary.

Data analysis in terms of age has changed

  1. As it stands, there are two sections numbered 3.6.

Answer: section numbering has been corrected

  1. The discussion section needs to draw more attention to what the results mean and what the implications are for this study – there is substantial repetition of results and no discussion at all about other correlates that are important to consider when assessing motor skills. For example, other factors that are associated with motor development include overweight/obesity, motor skill teaching, access to facilities etc. These factors have not been acknowledged and may explain some of the findings.

Answer: The discussion has been significantly rebuilt and supplemented. Some repetitions have been removed. Currently, we have focused on emphasizing what our research results mean and we have shown how other reflex researchers showed various types of relationships. Weight, height and BMI correlants were analyzed. Unfortunately, for the given group as a whole, we were unable to collect more information in common. Some of the children underwent additional tests, such as gait analysis or sensorimotor assessment of the child by the parent and teachers, which was described in other articles from the series of our research on reflexes, which I mentioned in the description of our PRACS project. 

Professional linguistic proofreading was also performed.

Kind regards 

Anna Pecuch

Reviewer 2 Report

The purpose of the current study was to assess three tonic reflexes and compare the presence of these reflexes with motor skill in a sample of 112 Polish children (aged 4-6). Generally, the idea of determining a possible correlation between primary reflexes and motor skill development is interesting. However, there is currently a missing link as to what the presence of these primary reflexes indicates and how the results of this study can be interpreted. Currently, it is unclear as to whether the presence of these reflexes’ is indicative of an underlying medical condition or if they are just an indicator of slower development and will disappear as the child gets older. Currently, there is no specific research question or hypothesis aimed at answering the research question. This has led to an unfocused narrative within the manuscript. Significant issues related to the statistical analysis and grammatical and typographical mistakes throughout the manuscript limit the ability to interpret the data presented. Specific comments are included below:

Major Revisions:

1.1 Statistics – Generally, the description of the statistics used in the current study are insufficient to meaningfully interpret the data as presented. As an example, it is unclear what is meant by the MOT results being “close to the normal distribution”. The data will either be normally distributed or not and the statistics performed will then correspond with that result. Since it appears the data is not normally distributed, would it not be most appropriate to use non-parametric test on this data?

1.2 Statistics - It is also unclear why were only the final MOT test result was used for analysis? Was this to limit effects of learning the task itself? If so, this should be addressed in the methods as to how the scores were recorded and analyzed.

1.3 Statistics and Line 206 – Since it appears the data is not normally distributed; it does not appear that a parametric test (ANOVA) is the appropriate test for the data described. Furthermore, it is not clear how this ANOVA was performed. What are the factors that are being analyzed in the ANOVA and how many comparisons are there in each factor? Providing specifics as to the test being performed would enhance the meaningfulness of the result. Are these separate 1 (APR or MOT) x 3 (Age 4,5,6) ANOVAs? Generally, ANOVA results are reported in a standard way (see below). Including all necessary information including degrees of freedom, F Statistic, and Significance level is the minimum information necessary.

(F(x,y) = 41.09, p <0.01) – x and y are the degrees of freedom

1.4 Statistics – An ANOVA is being performed but there is currently no mention of post-hoc tests to determine where differences occurred within the ANOVA. This is necessary to include both in the statistical analysis section and the results. Overall, it is not clear from the description what was tested, what is different, and how this can be interpreted. Importantly it is also not clear if an adjustment for multiple comparisons has been made in the data as presented. If so, which adjustment. If not, what is the reason?

2.1 Discussion Line 258 – A vital question is raised early in the discussion that is fundamental to the current work. The question is raised as to whether the occurrence of primary reflexes always constitute a serious problem for psychomotor development. Is the presence of these reflexes’ indicative of an underlying medical condition or are they just an indicator of slower delay that will disappear as the child gets older? This fundamental issue needs to be addressed in the introduction. Is the current study aimed at determining an answer to this question? A clear statement as to what information these primary reflexes provide is necessary or it needs to be incorporated into the study research question and testable hypothesis within the introduction.

3.1 Discussion - The discussion is currently unfocused and unclear as to the purpose of the paper and what research question is being tested and addressed. This is a fundamental issue with the structure of the paper and the research design. Critical information is omitted including a main research question and testable hypothesis which limits the direction and focus of the research question and manuscript. The discussion is presented mainly as a descriptive breakdown of the percentages of kids that have reflexes present, and how well they performed on the motor tasks compared to several other groups. It is currently unclear what conclusions can be drawn from the information presented.

4.1 Conclusions Line 363 - The conclusions presented do not appear to be based on data presented in the paper. Currently, the conclusions are general and provide little direction for future research. Focusing the intent of the article to a specific research question that is being tested with the methods employed would strengthen the ability to make definitive conclusions. Addressing issues related to the delay of integration of primary reflexes being correlates with lower motor scores is important. However, the link between assessing primary reflexes and improving functional outcomes for kids with targeted rehabilitation strategies is currently lacking. It is unclear how the cause of any of these movement disorders will be determined by screening primary reflexes and what rehabilitation strategies would be effective for improving motor skills, perceptual functions, and emotional functions.

Minor Revisions:

Methods - Are the same researchers performing all subjective clinical assessments on these children? What precautions have been taken to ensure standardization is occurring. Do the researchers testing reflexes also test motor skills? Is there potential for non-conscious bias if the tester is familiar with the other score? This information should be incorporated in the research design. If required, information related to this issue should be added as a limitation to the current study.

Line 75 – Change “63 grils” to “63 girls”

Line 86 – Change “reserchers” to “researchers”

Table 2 – Ensure all data points are aligned.

Table 3 – Table needs a clear organization and alignment. Currently unorganized. It is also not clear what the letters in brackets refer to.

Table 4 – Ensure all data points are aligned.

Results Line 165 – 171 – There are numerous typographical errors in this section which limit the readability.

Results – Only report significant figures.

Figure 7 Legend – Further description of this data is required. This should include information on what the values represent, how they were calculated and what the x-axis and y-axis are referring to.

Figure 8 legend – Similar to Figure 7, additional information is required to allow for clear interpretation of this data without having to refer to the methods to determine what the x and y axis are referring to.

Results line 224 – Change “higer” to higher

Within the manuscript gender and sex are used interchangeably within the manuscript. Was both sex and gender asked of each participant or is this a biological sex at birth?

Discussion Line 240 Replace “specyfic” with specific

Discussion Line 261 – Awkward sentence structure. Please reword.

Discussion Line 337 – Missing references/Format error

Author Response

Dear Reviewer

In the attachment I am sending the replies to your comments. Thank you very much for them because they have definitely contributed to the improvement of the quality of the manuscript.

Rev.2.: Currently, it is unclear as to whether the presence of these reflexes’ is indicative of an underlying medical condition or if they are just an indicator of slower development and will disappear as the child gets older.

Answer: After reading your comments and other reviewers, I also noticed that the introduction was too minimalistic by me. Currently, I have enriched it significantly with a description of the mechanisms that characterize individual reflexes and what they influence. I hope that now the connections between the occurrence of reflexes and the disruptions of specific skills are much more visible and understandable. I also partially moved the paragraph that talked about the occurrence of reflexes in the population of children, and what their occurrence, according to the authors of various studies, may influence. In the previous version, this paragraph was part of the discussion.

The presence of reflexes indicates neurotransmitter immaturity. The presence of one or two reflexes is common in preschool children and should certainly not be considered a symptom of disease or severe pathology.

In addition, temporary studies of preschool and school-age children indicate that with the development of the child, few and poorly expressed reflexes have a chance of being completely integrated (disappearing). However, when the number of reflexes and their degree are increased, they may significantly interfere with motor development.

Rev.2.: Currently, there is no specific research question or hypothesis aimed at answering the research question. This has led to an unfocused narrative within the manuscript.

Answer: Research question and hypotheses have been added.

Rev.2.: Significant issues related to the statistical analysis and grammatical and typographical mistakes throughout the manuscript limit the ability to interpret the data presented.

Answer: Linguistic and typographical errors have been corrected. A professional linguistic correction was also made.

  • Statistics – Generally, the description of the statistics used in the current study are insufficient to meaningfully interpret the data as presented. As an example, it is unclear what is meant by the MOT results being “close to the normal distribution”. The data will either be normally distributed or not and the statistics performed will then correspond with that result. Since it appears the data is not normally distributed, would it not be most appropriate to use non-parametric test on this data?

Answer: The distribution of the data was checked again in the statistical analysis. The data obtained in the MOT test are normally distributed, as indicated in the text. The data obtained in the study of reflexes are not normally distributed, which is also indicated in the text.

After consulting the statistician, we made a decision to change the tests to better suited to the research question and hypothesis posed.

  • Statistics - It is also unclear why were only the final MOT test result was used for analysis? Was this to limit effects of learning the task itself? If so, this should be addressed in the methods as to how the scores were recorded and analyzed.

Answer: Both the final result and its individual tests were used for the analysis. Analysis of the correlation of individual reflexes with the final result of the MOT test was also performed. Presenting the correlation of individual reflexes with individual tests gives too much data, which would be difficult to present in an accessible way for the reader. It is also not relevant to the research question: we do not want to show which reflex correlates with which test from MOT. Our aim was to show that in children with an increased index of the number and degree of expressed reflexes, motor skills related to balance, coordination and concentration abilities may be impaired. The methodological description describes how the reflexes were assessed, and how the individual MOT tests were assessed. In the description of the statistics it was stated that the correlation coefficient was calculated to compare these data.

1.3 Statistics and Line 206 – Since it appears the data is not normally distributed; it does not appear that a parametric test (ANOVA) is the appropriate test for the data described. Furthermore, it is not clear how this ANOVA was performed. What are the factors that are being analyzed in the ANOVA and how many comparisons are there in each factor? Providing specifics as to the test being performed would enhance the meaningfulness of the result. Are these separate 1 (APR or MOT) x 3 (Age 4,5,6) ANOVAs? Generally, ANOVA results are reported in a standard way (see below). Including all necessary information including degrees of freedom, F Statistic, and Significance level is the minimum information necessary.

(F(x,y) = 41.09, p <0.01) – x and y are the degrees of freedom

Statistics – An ANOVA is being performed but there is currently no mention of post-hoc tests to determine where differences occurred within the ANOVA. This is necessary to include both in the statistical analysis section and the results. Overall, it is not clear from the description what was tested, what is different, and how this can be interpreted. Importantly it is also not clear if an adjustment for multiple comparisons has been made in the data as presented. If so, which adjustment. If not, what is the reason?

Answer: After talking to the statistician and on the advice of reviewer # 1, the data was analyzed by age group using a multiple regression model .

2.1 Discussion Line 258 – A vital question is raised early in the discussion that is fundamental to the current work. The question is raised as to whether the occurrence of primary reflexes always constitute a serious problem for psychomotor development. Is the presence of these reflexes’ indicative of an underlying medical condition or are they just an indicator of slower delay that will disappear as the child gets older? This fundamental issue needs to be addressed in the introduction. Is the current study aimed at determining an answer to this question? A clear statement as to what information these primary reflexes provide is necessary or it needs to be incorporated into the study research question and testable hypothesis within the introduction.

Answer: The aim of the study was to show the existence of a relationship between an increased index of reflex activity and lower motor skills of children. The a multiple regression model has been shown that the older age of child and the lower reflex activity, the result obtained for the child's motor skills is higher. The presence of reflexes in older children is an indicator of slower neuromotor development, and in children with more and more reflexes, it may not be self-extinguishing.

3.1 Discussion - The discussion is currently unfocused and unclear as to the purpose of the paper and what research question is being tested and addressed. This is a fundamental issue with the structure of the paper and the research design. Critical information is omitted including a main research question and testable hypothesis which limits the direction and focus of the research question and manuscript. The discussion is presented mainly as a descriptive breakdown of the percentages of kids that have reflexes present, and how well they performed on the motor tasks compared to several other groups. It is currently unclear what conclusions can be drawn from the information presented.

Answer: The discussion has been significantly rebuilt and supplemented. Some repetitions have been removed.  Currently, we have focused on emphasizing what our research results mean and we have shown how other reflex researchers showed various types of relationships.

4.1 Conclusions Line 363 - The conclusions presented do not appear to be based on data presented in the paper. Currently, the conclusions are general and provide little direction for future research. Focusing the intent of the article to a specific research question that is being tested with the methods employed would strengthen the ability to make definitive conclusions. Addressing issues related to the delay of integration of primary reflexes being correlates with lower motor scores is important. However, the link between assessing primary reflexes and improving functional outcomes for kids with targeted rehabilitation strategies is currently lacking. It is unclear how the cause of any of these movement disorders will be determined by screening primary reflexes and what rehabilitation strategies would be effective for improving motor skills, perceptual functions, and emotional functions.

Answer: The conclusions were reformulated in terms of the answer to the research question. We hope that the information on reflexes added to the article and their relationship with motor disorders and what rehabilitation strategies can be taken to improve them (including the reflex extinction therapy in exercises - most of the information was provided in the discussion).

Minor Revisions

Methods - Are the same researchers performing all subjective clinical assessments on these children? What precautions have been taken to ensure standardization is occurring. Do the researchers testing reflexes also test motor skills? Is there potential for non-conscious bias if the tester is familiar with the other score? This information should be incorporated in the research design. If required, information related to this issue should be added as a limitation to the current study.

Answer: Information on researchers and standardization has been added to the text. Those who examined the children were entitled to do so through appropriate courses and training. During the studies, precautions were taken to minimize the risk of the result of one study having an influence on the other.

Line 75 – Change “63 grils” to “63 girls”         Answer: changed

Line 86 – Change “reserchers” to “researchers”        Answer: changed

Table 2 – Ensure all data points are aligned.

Table 3 – Table needs a clear organization and alignment. Currently unorganized. It is also not clear what the letters in brackets refer to. Answer: corrected

Table 4 – Ensure all data points are aligned.  Answer: corrected

Results Line 165 – 171 – There are numerous typographical errors in this section which limit the readability. Answer: corrected

Results – Only report significant figures Answer: on the recommendation of the reviewer No. 3, the symbol * p <0.05 was added for significant results

Figure 7 Legend – Further description of this data is required. This should include information on what the values represent, how they were calculated and what the x-axis and y-axis are referring to. Answer: corrected

Figure 8 legend – Similar to Figure 7, additional information is required to allow for clear interpretation of this data without having to refer to the methods to determine what the x and y axis are referring to. Answer: corrected

Results line 224 – Change “higer” to high er         Answer: changed

Within the manuscript gender and sex are used interchangeably within the manuscript. Was both sex and gender asked of each participant or is this a biological sex at birth?

Answer: The participants were not asked about it. The child's gender has been determined on the basis of the personal data provided by the parent in the information sheet.  Neither of them reported any differences in the child between their biological sex at birth and their current gender identity.

Discussion Line 240 Replace “specyfic” with specific              Answer: changed

Discussion Line 261 – Awkward sentence structure. Please reword.           Answer: changed

Discussion Line 337 – Missing references/Format error     Answer: changed

Reviewer 3 Report

This manuscript presents a relevant topic to publish in the Brain Sciences Journal, which could be accepted with some minor revisions. 

In my opinion, the introduction provides adequate information and structure to set up the research questions raised in the manuscript; the methodology provides sufficient detail, however, it should be improved; the results section is sufficiently clear and precise; the discussion of results based on previous literature.

After carefully reading your manuscript, I point out some aspects that must be improved and corrected:

- Many aspects of spelling and grammatical errors should be corrected. Please, correct what is pointed out in the body of the manuscript;

Several aspects of the methodological procedures should be described:

Introduce a brief psychometric characterization of the motor assessment instruments used (validity and reliability);

Who evaluated the children? a trained investigator?

Location of the evaluation? evaluation duration? each child did all the tests on the same day?

Authors must include the biographical reference/authorship when referring to motor tests for the 1st time in the body of the text (line 80-81);

- in table 2 and fig. 7, the authors should put the levels of reflex activity in full (low activity, high activity, maximum activity);

- in figures 7 and 8, put a label on the y-axis (% children?)

-  put a caption in tables 5 and 8 in relation to the abbreviations for a better reading of the data;

-  the significance levels should be marked with * and with the respective legend (eg * p‹0.05; ** p‹0.01; ***p‹ 0.001; )

 - in table 1, put in the age column the mean and the standard deviation of the age for each age group;

 - All statistical symbols must be in italics (N, n, p,  ....). 

Author Response

Dear Reviewer Thank you for your comments which improved the quality of our manuscript.

Many aspects of spelling and grammatical errors should be corrected. Please, correct what is pointed out in the body of the manuscript.

Answer: Linguistic and typographical errors have been corrected. A professional linguistic correction was also made.

-Introduce a brief psychometric characterization of the motor assessment instruments used (validity and reliability)

Answer: The information was added in the text.

-Who evaluated the children? a trained investigator?

Answer: Information on researchers and standardization has been added to the text. Those who examined the children were entitled to do so through appropriate courses and training.

-Location of the evaluation? evaluation duration? each child did all the tests on the same day?

Answer: Information abou location has been added to the text. The tests were performed on one day (with a break between the tests so that the child could rest) or on two separate days if the child showed signs of fatigue or distraction. The MOT test took about 20 minutes and the reflex test 10 minutes.

The research was conducted in three different facilities, including two kindergartens and a clinic. The children were usually examined on the same day, although if the child showed, for example, symptoms of fatigue after the MOT test, the reflex test was performed on a different day.

-Authors must include the biographical reference/authorship when referring to motor tests for the 1st time in the body of the text (line 80-81);

Answer: References have been addend

- in table 2 and fig. 7, the authors should put the levels of reflex activity in full (low activity, high activity, maximum activity);

Answer: it has been changed

- in figures 7 and 8, put a label on the y-axis (% children?)

Answer: it has been changed

-  put a caption in tables 5 and 8 in relation to the abbreviations for a better reading of the data

Answer: it has been changed

-  the significance levels should be marked with * and with the respective legend (eg * p‹0.05; ** p‹0.01; ***p‹ 0.001; )

Round 2

Reviewer 1 Report

Dear authors

Well done on a much improved paper. There are a number of things that need to be addressed:

  1. Autism spectrum disorder and cerebral palsy are not intellectual disabilities (as implied in section 2.1).
  2. In tables 1, 5, 6, 7, 9, and 11 there appears to be no consistency with decimal points within each individual table.
  3. The first (new) paragraph of the discussion has no context and is a repeat of the results.
  4. In the discussion, cerebral palsy, autism spectrum disorder and stroke are defined as having 'highly compromised neurology' - this contradicts other definitions in the paper, but is also not entirely correct terminology. 
  5. The discussion is substantially longer but does not articulate a clear message and needs to be restructured to make sense to a reader. It would be helpful to structure it to first explain the main findings, then explain how each finding compares to existing literature, and then explain the contribution of this manuscript to the evidence base.
  6. There are parts of the discussion and the conclusion that now mention physical activity and physical activity recommendations are stated, but the paper does not present results or discussion that support the link between reflex activity, motor skills and physical activity.

Author Response

Dear Reviewer 1

Thanks for your comments. I hope that the changes made by me are sufficient. If changes still need to be made, it would be helpful if I received comments made on the manuscript.

1. Autism spectrum disorder and cerebral palsy are not intellectual disabilities (as implied in section 2.1).

Answer: Has been corrected

  1. In tables 1, 5, 6, 7, 9, and 11 there appears to be no consistency with decimal points within each individual table.

Answer: Has been corrected. Means and standard deviations as well as percentages are given to one decimal place, and p to three decimal places.

  1. The first (new) paragraph of the discussion has no context and is a repeat of the results.

Answer: Perhaps I have not clearly indicated which results this paragraph refers to. The paragraph concerns the results presented in tables 8 and 9 of the results (correlation of reflexes with the results of the MOT test), and thus it refers to the most important research question posed in the article. I did not delete this paragraph, but edited it and expanded it a bit, as Reviewer 2 believes that it is the most important part of the discussion (that's why it was placed at the beginning). The discussion is not a chronological overview of the results, the issues discussed in it are ranked according to the importance of issues and results in the context of the research questions posed. To make it easier for the reader to navigate the article, paragraph subtitles in discusion have been added. Also, as suggested by the second reviewer, it was divided into two parts.

  1. In the discussion, cerebral palsy, autism spectrum disorder and stroke are defined as having 'highly compromised neurology' - this contradicts other definitions in the paper, but is also not entirely correct terminology.

Answer: Has been corrected

  1. The discussion is substantially longer but does not articulate a clear message and needs to be restructured to make sense to a reader. It would be helpful to structure it to first explain the main findings, then explain how each finding compares to existing literature, and then explain the contribution of this manuscript to the evidence base.

Answer: The present ones have divided the discussion into subsections which I have appropriately named so that they become more accessible to the reader. In the comments, I indicate  where I discuss the results in relation to the current state of knowledge and what our research brings new.

  1. There are parts of the discussion and the conclusion that now mention physical activity and physical activity recommendations are stated, but the paper does not present results or discussion that support the link between reflex activity, motor skills and physical activity.

Answer: In the article, we focused mainly on presenting the results of correlation of reflexes activity with motor skills and discussing these relationships (first paragraph). Perhaps unnecessary confusion was introduced by the alternate use of the term motor performance and physical fitness - this has been eliminated from the text. In the paragraph talking about the possibility of performing exercises that integrate the activity of reflexes, I meant specifically directed motor activity (therapeutic exercises).

Are the conclusions supported by the results?

The conclusions have been reformulated so that the first part is the answer to the research question and hypotheses, and the second part is a guide for practitioners.

A linguistic correction was also made.

Reviewer 2 Report

The authors are to be commended for their restructuring of the article. Clarifying the purpose and hypothesis have allowed for a more directed discussion. The changes made to the statistics are clear and justified. The conclusions are justified by the purpose and methods and are tied to the results of the study. There are a few minor clarifications to be addressed that are listed below.

Minor Limitations:

  • Introduction Line 34 - Sentence structure should be improved. Is this supposed to read “Primary reflexes are stereotypical and involuntary motor reactions governed by the brainstem develop during fetal life.”
  • Introduction Line 89 – 91 – This sentence clarifies the idea but the term “or not pace” still leads to confusion. Please clarify this wording so the idea is clear.
  • Results Line 338 – Change “no” to “not”
  • Discussion Line 350 – Remove “her”
  • Discussion Line 348 – 359 – This is a valuable addition. It may be valuable to highlight the novelty of the findings in the current study to emphasize the unique components and relevance to the previous literature. Breaking this first part of the discussion into a separate paragraph may also improve readability.
  • Discussion Line 432 – “Reflexes” is repeated two words in a row. Altering sentence structure would make the point clearer.
  • Discussion Line 483 – Please clarify – This is an issue that “should not” or “should” be addressed in a separate article?

Author Response

Dear Reviewer 2

Thanks for your comments. 

Introduction Line 34 - Sentence structure should be improved. Is this supposed to read “Primary reflexes are stereotypical and involuntary motor reactions governed by the brainstem develop during fetal life.”

Answer: The sentence has been corrected.

Introduction Line 89 – 91 – This sentence clarifies the idea but the term “or not pace” still leads to confusion. Please clarify this wording so the idea is clear.

Answer: The sentence has been corrected.

Results Line 338 – Change “no” to “not”

Answer: The sentence has been corrected.

Discussion Line 350 – Remove “her”

Answer: Removed

Discussion Line 348 – 359 – This is a valuable addition. It may be valuable to highlight the novelty of the findings in the current study to emphasize the unique components and relevance to the previous literature. Breaking this first part of the discussion into a separate paragraph may also improve readability.

Answer: At the beginning of the discussion, information about the novelty in the approach to the subject of reflexes was added. The first paragraph has been split.

Discussion Line 432 – “Reflexes” is repeated two words in a row. Altering sentence structure would make the point clearer.

Answer: The sentence has been corrected.

Discussion Line 483 – Please clarify – This is an issue that “should not” or “should” be addressed in a separate article?

Answer: The sentence has been corrected.

A linguistic correction was also made.
